# Magnetic Material in Triboelectric Nanogenerators: A Review

**DOI:** 10.3390/nano14100826

**Published:** 2024-05-08

**Authors:** Enqi Sun, Qiliang Zhu, Hafeez Ur Rehman, Tong Wu, Xia Cao, Ning Wang

**Affiliations:** 1Center for Green Innovation, School of Mathematics and Physics, University of Science and Technology Beijing, Beijing 100083, China; d202310453@xs.ustb.edu.cn (E.S.); d202110424@xs.ustb.edu.cn (Q.Z.); hafeezurrehmandgk59@gmail.com (H.U.R.); 2National Institute of Metrology China, Beijing 100029, China; wut@nim.ac.cn; 3Beijing Institute of Nanoenergy and Nanosystems, Chinese Academy of Sciences, Beijing 100083, China

**Keywords:** triboelectric nanogenerator, magnetic material, self-powered sensing, harvest energy

## Abstract

Nowadays, magnetic materials are also drawing considerable attention in the development of innovative energy converters such as triboelectric nanogenerators (TENGs), where the introduction of magnetic materials at the triboelectric interface not only significantly enhances the energy harvesting efficiency but also promotes TENG entry into the era of intelligence and multifunction. In this review, we begin from the basic operating principle of TENGs and then summarize the recent progress in applications of magnetic materials in the design of TENG magnetic materials by categorizing them into soft ferrites and amorphous and nanocrystalline alloys. While highlighting key role of magnetic materials in and future opportunities for improving their performance in energy conversion, we also discuss the most promising choices available today and describe emerging approaches to create even better magnetic TENGs and TENG-based sensors as far as intelligence and multifunctionality are concerned. In addition, the paper also discusses the integration of magnetic TENGs as a power source for third-party sensors and briefly explains the self-powered applications in a wide range of related fields. Finally, the paper discusses the challenges and prospects of magnetic TENGs.

## 1. Introduction

Given the continuous growth of the global energy demand, people are increasingly concerned about the sustainability and renewability of energy [1,2]. The limited nature of traditional energy sources and their impacts on the environment make the development of new energy sources a top priority [3]. In the pursuit of new energy sources, TENGs have emerged as one of the highly anticipated energy-converting technologies on the basis of the coupled effects of triboelectric charging and electrostatic induction [4,5]. An interaction between two distinct frictional electrode materials results in the generation of frictional charges on their surfaces, as well as the conversion of mechanical energy into electrical potential in the event of additional electrostatic induction. TENGs can convert small mechanical energies from human body movements [6,7,8,9,10,11,12], breezes [13,14,15], rotations [16], sound waves [17,18,19], water waves [20,21,22,23,24,25,26,27,28], and other sources into electrical energy for various wearable electronics [29].

TENGs possess high flexibility and portability, allowing them to be embedded into various devices and systems, and providing them with self-sustaining power sources. This technology holds great potential for applications in precision instruments [30,31,32,33], wearable devices [34,35,36], chemical reactions [37,38,39], sensors [40,41,42,43], and other portable personal electronic products [44,45,46]. It not only provides continuous and stable energy for these devices but also has the potential to reduce reliance on traditional batteries to some extent, thereby minimizing electronic waste generation and promoting environmental friendliness [47,48]. Therefore, as an innovative energy technology, TENGs offer a more sustainable and eco-friendly energy solution for humanity [49,50,51]. Science and technology are thought to have the potential to play a significant role in the future energy sector, providing our intelligent society with more convenience and sustainability as a result of ongoing advancements and the growth of their applications.

At present, magnetic materials are showing their potential for the design of high-performance TENGs due to their high magnetic conductivity, stability, adjustable magnetism, and controllable magnetic fluid properties [52,53,54]. Adding magnetic materials to TENGs improves their energy conversion effectiveness, stability, and longevity while also expanding their controllability and adaptability [55,56,57,58,59].

Currently, magnetic materials, including neodymium iron boron (NdFeB), nickel ferrite (NiFe_2_O_4_), and iron oxide, can be used in TENGs to harness mechanical energy from sources such as wind [55,56], magnetic fields [57], finger bending [58], water flow [59], and compression [57]. These TENGs have been extensively studied in various applications, including self-powered sensors [60], robots [61], flexible wearable devices [62], and self-powered integrated systems [63]. Moreover, the variation in frictional electric output induced by mechanical or magnetic stimulation can be directly utilized as sensing signals, thereby introducing self-powered sensors based on TENGs. Therefore, self-powered intelligent systems can be developed in various fields, including the Internet of Things [64,65], flexible devices [66], drug delivery [67], and self-propelled robots [61].

On the basis of predecessors, this paper more fully summarizes the application of magnetic materials in TENGs and self-powered sensors. The investigated magnetic materials include a variety of soft ferrites, hard magnetic materials, and composite magnetic materials. In addition to discussing the most promising options at the moment and outlining new strategies to build more intelligent and multifunctional magnetic TENGs, we also review the current state of development of magnetic TENGs and prospects for enhancing their energy conversion performance. Additionally, this article explores the incorporation of TENGs as power sources for external sensors and provides a brief overview of the self-powering applications in various related fields, as depicted in Figure 1. Finally, the challenges and prospects of applying magnetic materials in TENGs are discussed.

## 2. Fundamentals of Triboelectricity and Magnetic Interactions in TENGs

The movement of the conductor in the magnetic material’s field causes electromagnetic induction when magnetic materials are employed as the friction material in a TENG, which can improve the TENG’s performance. Therefore, the triboelectric effect and the electromagnetic inductive phenomenon are produced when magnetic materials are added to the TENG.

### 2.1. Triboelectric Effect

The triboelectric effect relies on the combination of static induction and frictional charging, enabling the effective conversion of small mechanical energy into usable electrical energy. A TENG is made up of two distinct friction materials, typically chosen to have a large difference in electronegativity. The frictional forces between these two distinct materials cause electrons to move from one electrode material to the other when they come into contact and then separate. Mechanical energy can be converted into electrical energy through the creation of an electric current by the potential difference created by the transfer of charges between the two electrode materials [68]. Therefore, through periodic friction, a significant amount of electric current can be generated, which can be applied in various fields [69], such as the Internet of Things [70,71,72], self-powered sensors [73,74,75,76], self-driving systems [33,77,78,79], and flexible electronic devices [80,81,82,83].

### 2.2. Electromagnetic Induction

Faraday’s electromagnetic induction is the term used to describe the phenomenon known as electromagnetic induction, which occurs when an electric current is induced in a conductor as a result of the interaction between electric and magnetic fields [84]. There are two occurrences of electromagnetic induction. First, an induced electromotive force will be generated by the closed circuit when the magnetic field shifts about it. The pace at which the magnetic field changes determines how much of an induced electromotive force there is. The second occurrence is when a conductor moves relative to a magnetic field, which also generates an induced electromotive force in the conductor. The conductor’s motion velocity and the magnetic field’s intensity both affect the strength of this produced electromotive force. Electromagnetic induction has significant applications in technologies such as generators, transformers, and induction heating. An induced electromotive force is produced in generators by mechanically moving conductors via a magnetic field; this force is then transformed into electrical energy [85]. In transformers, the transmission and transformation of alternating current are also based on the principles of electromagnetic induction. Induction heating utilizes the characteristic of generating heat in a conductor through induced currents and is used for heating metallic objects. All these applications rely on the fundamental principles of electromagnetic induction.

### 2.3. Working Modes of TENGs

As seen in Figure 2, TENG has been divided into four fundamental operating modes since its development in 2012, according to the direction of friction and its structure: contact-separation mode (CS), lateral sliding mode (LS), single-electrode mode (SE), and freestanding triboelectric-layer mode (FT). Each of these four basic working modes has its advantages and characteristics in harvesting mechanical energy under different environmental conditions.

Firstly, the CS TENG operates when two electrode materials in the vertical direction experience forces that induce opposite movements. Due to the frictional forces on the surfaces, the properties of the materials cause charge transfer when the surfaces separate or come into a connection, leading to the separation of charges and the development of a potential difference (Figure 2a). When an external circuit is connected to a load, current flows out [86]. By undergoing periodic contact and separation, this mode can generate a continuous current signal. Due to its straightforward structural design and extensive research, this mode is appropriate for low-frequency energy harvesting.

Secondly, the LS TENG has a similar structure to the vertical CS. When two opposing electrode materials are subjected to external forces, relative displacement occurs on the surfaces of the electrode materials. Surface friction charges are produced as a result of continuous forces that periodically cause the electrode materials to come into contact and separate (Figure 2b). To harvest energy, the current is output to an external circuit via the potential difference created between the electrodes by the accumulated charges [87]. The sliding friction mode is suitable for high-frequency energy harvesting and requires the selection of friction-resistant materials to reduce losses and improve stability.

Next is the SE. Similar to the vertical CS, external forces cause the electrode materials to undergo opposite movements in the vertical direction (Figure 2c). A reference electrode is one of the electrode materials; charge transfer is induced and a potential difference is generated by the contact and separation of the material surfaces, which transfers energy to an external circuit [88]. In the SE TENG, the moving part can be disconnected from the electrode, simplifying the structural design but reducing the efficiency of energy conversion.

Lastly, the FT involves the relative motion and friction-induced charge transfer between dielectric and electrode materials. Current flows between the electrodes once a potential difference has formed. The movement of the dielectric layer controls the direction of current flow (Figure 2d). The freestanding mode allows for the mixed collection of various forms of energy but increases the complexity of the structure [89]. 

## 3. Types of Magnetic Materials and Their Applications in TENGs

Based on their magnetic properties and qualities for specific applications, different types of magnetic materials can be categorized as hard, soft, or magnetic composite materials. Different magnetic materials also play different roles in TENGs. Magnetic materials are essential for electromagnetic generators (EMGs), and they can also be used as materials for hybrid nanogenerators (HNGs) by combining EMGs and TENGs [90]. The following section introduces the characteristics of different magnetic materials and their research progress in TENGs.

### 3.1. Hard Magnetic Materials in TENG

#### 3.1.1. Hard Magnetic Materials

Hard magnetic materials, often called permanent magnetic materials, refer to materials that can maintain their magnetism for a long time after being magnetized (Table 1). They typically have high magnetization intensity. In TENGs, hard magnets mainly consist of different grades of NdFeB permanent magnets. Given their ability to transform electrical energy into mechanical energy, permanent magnets are highly sought after as crucial industrial materials in a variety of applications, including electric motors [91], data storage devices [92], and magnetic actuators [93].

#### 3.1.2. Application of Hard Magnetic Materials in TENGs

Hard magnetic materials efficiently convert weak mechanical energy into electrical energy for load usage in nanogenerators. They can be divided into non-contact and contact varieties according to whether the nanogenerator and the mechanical motion source are directly touching. In 2012, Cui et al. developed two types of non-contact magnetic-driven nanogenerators using electrospinning: one utilizes radial bending of nanowires for power generation, and the other utilizes axial bending of nanowires [94]. The feasibility of non-contact magnetic-driven nanogenerators was demonstrated for the first time by driving the friction layer to produce mechanical motion using NdFeB and Fe_3_O_4_ permanent magnets and employing a single zinc oxide (ZnO) microfiber (Figure 3a). Additionally, a non-contact working mode was achieved by integrating an array of electrospun lead zirconate titanate (PZT) nanowires into the nanogenerator. This nanogenerator exhibited an output voltage (Voc) of 3.2 V and an output current of 50 nA (Figure 3b,c). The maximum output power density of the device reached 170 μW/cm^3^. By successfully lighting an light-emitting diode (LED) screen, the nanogenerator paved the way for the use of nanogenerators for magnetic sensing and non-contact mechanical energy collection and utilization.

In order to construct a drug delivery system (DDS) for cancer treatment within the human body, Zhao et al. developed a unique magnetic TENG (MTENG) using magnetron sputtering technology [67]. The output performance of the TENG, reaching up to 70 V, was successfully improved by utilizing a permanent magnet to facilitate cyclic motion, causing the two friction layers to contact and separate. By utilizing a unique structural design, this contact-type MTENG ensures high and stable electrical output upon encapsulation and implantation, doing away with the requirement for traditional spacers. During the study, red blood cells (RBCs) loaded with the anticancer drug doxorubicin (DOX) were used as drug carriers (Figure 3d). The release of DOX from the RBC membrane is slow and controlled, but increases significantly upon stimulation by the MTENG, returning to normal levels after stimulation. This establishes a controllable DDS. Both in vivo and in vitro, the MTENG can regulate the DDS to provide low-dose DOX-induced cytotoxicity against cancer cells. These results demonstrate the significant therapeutic potential of magnetically driven TENGs in medical treatments and suggest broad applications in clinical settings.

In order to capture magnetic energy from transmission lines, Jin et al. devised and created a unique structured TENG based on magnetic balls (NdFeB) [95]. The central element of the TENG, the magnetic sphere, generates centrifugal force that drives the spring to perform a contact separation motion with the electrode (Figure 3e). Even in the presence of shocks from high currents, this TENG maintains constant output performance and can be activated in magnetic fields produced by gearbox currents greater than 400 A (Figure 3g). A maximum output of 1.5 kV, 20 µA, and 6.76 mW (Figure 3h) was attained on a 4 cm diameter spherical shell and a 12 mm diameter magnetic ball. Moreover, the diameter of the sphere shell directly controls the operating frequency of this TENG unit, enabling it to accommodate a broad spectrum of transmission currents. It has also successfully powered a wireless alarm system that monitors the temperature and inclination of electricity transmission cables.

Using 3D printing technology, a magnetic capsule TENG (MC-TENG) was created by Jiao et al. to harvest energy during weak mechanical actions [96]. The capsule structure is positioned between driving magnets in a fixed frame as part of the TENG’s architecture (Figure 3f). Current is generated in an independent frictional–electric layer mode by the oscillation-triggered magnetic forces driving the dielectric-encapsulated TENG. The magnet (conductivity: 625,000 Siemens/m) is used to periodically move the magnetic capsule, enhancing the durability of the TENG device. To find out how well the MC-TENG performed electrically in three different energy harvesting modes under cyclic loads, experiments and numerical analysis were carried out. The findings show that the capsule TENG structure affects the MC-TENG’s ability to harvest energy. With a closed circuit resistance of 10 GΩ and a maximum voltage range of 4 V, the copper MC-TENG system was determined to be the most efficient design. The MC-TENG concept offers a practical method for obtaining electrical energy from oscillations with low frequency and low amplitude, like waves in the ocean. 

In order to capture rotational energy, Bai et al. presented a unique jump-type TENG called the Spin-Toggle Rotating TENG (ST-RTENG), which combines a magnetic-coupled and buckling bistable mechanism [16]. When the buckling bistable beam is excited by a magnetic field, the magnet (NdFeB-N50, flux density: 1.2 T, permeability: 1.256 × 10^−6^ H/m, magnetization intensity: 3 × 10^6^ A/m) produces convex–concave leaps that improve the electrical output by increasing the contact force between the TENG functional materials. The driving component and the electromechanical conversion component can be physically separated using a magnetic-coupled non-contact energy transfer mechanism, which increases the system’s stability under challenging conditions (Figure 4a). The system’s coupled electromechanical dynamic model was created and experimentally verified. The outcomes show that the TENG’s performance is greatly improved by the magnetically coupled buckling bistable mechanism. At various rotational speeds, the best output performance can be attained by varying the magnet configurations and the smallest center distance (d). With two pairs of oppositely polarized magnets, the ST-RTENG may rotate at 150 r/min to reach a maximum voltage of 1235 V, and at 800 r/min to reach a maximum average power of 778 μW (Figure 4b). Under low-speed stimulation, the prototype can light up more than a thousand LEDs and provide wireless transmission and self-powered temperature monitoring.

Utilizing 3D printing technology, an Alternating Magnetic Field-enhanced TENG (AMF-TENG) based on Ampere’s force was presented by Zhang et al. to efficiently gather low-speed flow energy and enable self-powered sensing [55]. With the use of linked coils, frictional electrification, magnetic induction of magnets, and electrostatic induction effects, the AMF-TENG low-speed wind energy harvesting technique allows the TENG to capture wind energy at speeds as low as 1 m/s. Under the operation of an external magnetic field (rubidium magnets, magnetic field strength: 243 mT), the coils produce induced currents that drive the vertically connected and separated TENG (Figure 4d) by producing Ampere forces as a result of the current’s magnetic effect and the interaction with the external magnetic field. Furthermore, as the rotational speed decreases, the current flowing through the coils also does, resulting in less resistance for the AMF-TENG while rotating at low speeds. This increases the wind speed range for TENG applications by allowing the gathering of low-speed wind energy (1 m/s). At wind speeds of 1 to 5 m/s, the AMF-TENG’s Voc ranges from 20.9 V to 179.3 V (Figure 4c). The voltage of the AMF-TENG stays at roughly 92.5% after operating continuously for 100 thousand cycles, showing good output performance and stability. AMF-TENG can charge a 330 μF capacitor to 1.5 V in 316 s at a wind speed of 1 m/s (Figure 4e). This allows low-speed wind energy to be collected in natural settings and powers wireless light intensity sensors, agricultural temperature and humidity sensors, and other devices. This offers a fresh approach to capturing wind energy at low altitude and low speed.

A triboelectric-electromagnetic hybrid generator (TEHG) utilizing 3D printing technology to collect wind energy was proposed by Zhao et al. for self-powered IoT sensor nodes [56]. Wind energy can be transformed into electrical energy by combining the energy conversion techniques of TENGs and EMGs. Under the action of the magnetic field (NdFeB-N35), the magnetic flux of the moving coil constantly changes to generate the induced current and increase the output performance. This increases the output voltage range and significantly shortens the boosting time. The analysis’s findings demonstrate that when the external wind speed rises, the TENG’s output voltage and current also rise, demonstrating steady, high output performance. However, on the other hand, the effect becomes less significant after the voltage reaches 171 V with increasing wind speed. The TENG and EMG had the highest average power output values of 0.33 mW and 32.87 mW, respectively (Figure 4f,g), when the wind speed is 9 m/s. The respective ideal load resistance in this situation are 1.25 kΩ and 12 kΩ, respectively. After running for two seconds, the TEHG can power a temperature and humidity sensor or illuminate 200 serial LEDs. Using rectification and voltage regulation circuits, the HNG can power wireless environmental monitoring systems and provide data to smartphones via Bluetooth communication. This demonstrates the practical value and promotes the significance of TEHGs for the development of IoT systems.

Yang and colleagues presented a cantilever beam-based hybrid piezoelectric/triboelectric nanogenerator (HP/TENG) that makes use of ambient stray magnetic fields [97]. Lead zirconate titanate (PZT) serves as the piezoelectric material in this hybrid nanogenerator, which allows for the steady and effective collection of electromagnetic energy. The cantilever beam of the generator is made of a titanium (Ti) strip, and the frictional materials are polydimethylsiloxane (PDMS) and aluminum foil. NdFeB (N52) causes the cantilever beam to oscillate in an alternating magnetic field and produce periodic mechanical motion. The HP/TENG is powered by a 40 Oe magnetic field from the Helmholtz coil and can achieve short-circuit current (Isc) of 375 A, high open-circuit voltage (176 V), and an optimum output power of 4.7 mW (Figure 4h) thanks to the synergistic impact of triboelectric charging and piezoelectric polarization. Under optimal matching impedance, the HP/TENG generates approximately 2.08 times more total power than the piezoelectric nanogenerator (PENG) and 2.14 times more than the TENG when both mechanisms are operating. Moreover, this device can power humidity sensors and commercial wireless temperature sensors sustainably when combined with a circuit, indicating the possible uses for self-powered wireless sensing systems.

**Table 1 nanomaterials-14-00826-t001:** Applications and properties of hard magnetic materials in TENGs.

Material	Driving Mode	Key Materials	Output Voltage	Output Current	Output Power/ Power Density	Ref.
NdFeB/Fe_3_O_4_	No contact	PZT nanowires	3.2 V	50 nA	170 μW/cm^3^	[94]
Permanent magnet	Contact	PTFE/Ti	70 V	0.65 μA	—	[67]
NdFeB-N35	Contact	Nylon/FEP	1.5 kV	20 µA	6.76 mW	[95]
Permanent magnet	No contact	PLA	4 V	550 nA	300 nW	[96]
NdFeB-N50	No contact	FEP/Cu	1235 V	—	39.55 W/m^3^	[16]
Rubidium magnets	Contact	Poly lactic acid/PTFE	179.3 V	11.3 µA	0.66 mW	[55]
NdFeB-N35	Contact	Nylon/PTFE	171 V	14.6 µA	0.33 mW	[56]
NdFeB-N52	No contact	PDMS/Al	176 V	375 μA	4.7 mW	[97]

### 3.2. Soft Magnetic Materials in TENG

#### 3.2.1. Introduction to Soft Magnetic Materials

Soft magnetic materials refer to materials that cannot maintain magnetism for a long time after magnetization. Soft magnetic materials have very small residual magnetism and coercivity (less than 1000 A/m), and ferrite is the main soft magnetic material used in TENGs (Table 2). Soft magnetic materials have a wide range of applications in sensors [98,99], transformers [100], relays [101], and motors [102].

#### 3.2.2. Application of Soft Magnetic Materials in TENGs

Nanogenerators can benefit from the same soft magnetic material properties that are used in electronics. For example, Li fabricated a single-electrode TENG (S-TENG) using magnetron sputtering [62]. A dense nickel oxide forms on the surface of the metal nickel, which protects the electromagnetic properties of nickel. The good conductivity of nickel oxide and electric energy converted by magnetostatic energy improve the output characteristics and power generation efficiency of the TENG. With an open-circuit voltage, short-circuit current density, and transferred charge of roughly 80 V, 4 μA/cm^2^, and 26 nC (Figure 5a–c), respectively, and good stability, the ferromagnetic nickel-based material performs about 30% better than the copper-based S-TENG. Additionally, it only takes 265 s to charge a 100 μF capacitor to 2 V using this high-performance and stable ferromagnetic nickel-based electrode. This electrode has promising potential in the realm of self-powered electronic devices and has been used in flexible energy harvesters.

Then, Paralı et al. [103] successfully fabricated a nanogenerator based on polyvinylidene fluoride (PVDF) and nickel ferrite (NiFe_2_O_4_) using electrospinning (Figure 5d). The magnetoelectric, piezoelectric, and dielectric characteristics of the nanogenerator were altered by varying the weight ratios of NiFe_2_O_4_. The findings indicated that the nanogenerator containing 10 weight percent NiFe_2_O_4_ was more efficient than the pure PVDF fiber-based nanogenerator at 10 Hz. Furthermore, when exposed to a resistive load of 750 KΩ, the nanogenerator containing 7 wt% NiFe_2_O_4_ produced an ultimate voltage of 17.45 mV and a maximum power output of 0.40 μW. Furthermore, Durga and Hemalatha used electrospinning to create zinc ferrite (ZnFe_2_O_4_) filaments with a median width of 340 nm [104]. PVDF/ZnFe_2_O_4_ composite films were successfully created by loading various mass fractions of ZnFe_2_O_4_ fibers into a PVDF matrix. The material containing 15 wt% ZnFe_2_O_4_ was found to exhibit a high dielectric constant, ferroelectricity, and piezoelectricity, as confirmed by functional and structural group analyses. In addition, Figure 5e,f shows that the nanogenerator could achieve a maximum open-circuit voltage of 7 V with a force of 1.5 N and an output power of 4 mW with a load resistance of 500 kΩ.

In turn, Nawaz et al. investigated an HNG based on zinc ferrite-based nanocomposites [105], which is composed of three distinct kinds of nanogenerators: a TENG, PENG, and EMG (Figure 5g). ZnFe_2_O_4_ Magnetic nanoparticles (magnetization force: 15 kOe, magnetic flux density: 4.3 emu/g) increase the output performance of TENG by increasing the magnetic flux. The incorporation of zinc ferrite material enhances the piezoelectric response, triboelectric effect, and electromagnetic induction of the nanogenerators. The integrated HNG generated 30 V of Voc and 3.5 μA of output current (Figure 5h,i). Furthermore, the device’s sturdy and uncomplicated design, in conjunction with the nanocomposite’s multifunctionality for multimodal energy harvesting, showcases the potential of hybrid nanogenerators in the future. 

**Table 2 nanomaterials-14-00826-t002:** Applications and properties of soft magnetic materials in TENGs.

Material	Driving Mode	Key Materials	Output Voltage	Output Current	Output Power/ Power Density	Ref.
Ferroma-gnetic Ni	Contact	Ni-PDMS/Cu-PDMS	80 V	8 µA	6500 mW/m^2^	[62]
NiFe_2_O_4_	Contact	PVDF	17.45 mV	—	0.40 μW	[103]
ZnFe_2_O_4_	Contact	PVDF	7 V	—	4 mW	[104]
ZnFe_2_O_4_	Contact/No contact	PET/PVDF	58 V	6.2 µA	—	[105]

### 3.3. Magnetic Composite Materials in TENGs

#### 3.3.1. Hybrid Materials Combining Magnetic and Triboelectric Properties

Magnetic composites refer to composite materials with magnetic functionality that are formed by combining thermosetting or thermoplastic polymers with magnetic materials (Table 3). Common magnetic composites include magnetic epoxy resins, magnetic phenolic resins, magnetic polyethylene, magnetic alloys, and composite magnetic polymers [106,107,108].

#### 3.3.2. Synergistic Effects on TENGs through Composite Materials

Magnetic composites can expand the application field of raw materials and improve their performance in TENGs by combining different types of materials. For example, Qin et al. suggested a sliding triboelectric sensor with a magnetic array to enable real-time gesture interaction between robotic and human hands [58]. The rotational angle is proportionate to the tensile displacement at the joint (which serves as a fulcrum) when the finger bends (Figure 6a). Consequently, the sliding displacement can be used to calculate the degree of bending. The friction sensor works on the basic idea of inducing positive or negative pulses during finger traction motion (extension/flexion) and then counting the pulses per unit of time to determine the degree, speed, and direction of finger motion in real time. By limiting the sliding path and converting sliding motion into contact separation, the magnetic array-assisted sliding structure can increase the low-speed signal amplitude, stability, and durability. This study expands the potential for widespread applications by presenting an optimal approach based on wearable TENG sensors for real-time gesture interaction.

In addition, Ren et al. presented a novel magnetic-driven indirect electromagnetic-triboelectric HNG whereby electrostatic spinning was used to embed iron oxide (Fe_3_O_4_) nanoparticles as the triboelectric layer into a PVDF fiber membrane [109]. By utilizing the magnetic responsive properties of the triboelectric material, a non-contact driving contact separation mode was achieved using a magnet as the trigger, and the EMG was driven through the coupling between the magnet and the copper coil (Figure 6b). Excellent output performance and charging stability were displayed by the hybrid nanogenerator, which qualified it as a viable source of power for portable electronics and as a means of charging devices storing energy. Additionally, an entirely novel HNG prototype was presented, along with its possible use in creating an autonomous system.

Zheng et al. used tadpole-shaped CNTs@Fe_3_O_4_ nanoparticles using a hydrothermal reactor as the conductive filler and self-foaming polyurethane (PU) as the elastic matrix to create a stretchy combined foam-based TENG (CF-TENG) with adjustable microwave absorption (MA) capabilities [110]. The cation of Fe_3_O_4_ enhances the electrical conductivity and dielectric properties of the CNTs@Fe_3_O_4_ composites. The distinct morphology of CNTs@Fe_3_O_4_ nanoparticles is purposefully designed to improve MA capabilities and triboelectric output at the same time (Figure 6d). Therefore, the CF-TENG successfully harvested mechanical energy and lit up 35 commercial LEDs with an optimal power output of 147.9 μW (power density of 1.3 μW/cm^2^) under the vertical CS (Figure 6e). Additionally, the CF-TENG demonstrated exceptional MA performance, with a maximum tensile stress of 3.4 MPa, an effective frequency bandwidth of 4.37 GHz, a thickness of 2.55 mm, and a significant absorption intensity of −68.5 dB at 253 K (Figure 6c). Because of its exceptional performance, the CF-TENG is a viable option for electromagnetic protection in undetectable hostile settings and the harvesting of environmental mechanical energy.

A magnetically assisted indirect encapsulated TENG was presented by Huang et al. to gather blue energy and wind energy in high humidity settings [59]. The magnetic composite material provides the mechanical motion of contact separation for the TENG under the action of external forces. The Fe-Co-Ni blend that makes up the magnetic reaction layer was encapsulated to separate it from the active layer of the TENG, which is composed of PDMS, effectively isolating the device from the high humidity environment (Figure 6f) and ensuring stable TENG performance. For instance, when the relative humidity increased from 10% to 70%, the Voc of TENG decreased to 122 V, and when the relative humidity further increased to 90%, the Voc decreased to approximately 112 V, resulting in a 64% reduction. In contrast, the encapsulated magnetically assisted non-contact TENG performed more steadily, maintaining a voltage of about 300 V and only slightly deviating from that due to measurement (Figure 6g). Therefore, encapsulation is necessary for energy harvesting under high humidity conditions in practical applications. Moreover, the four novel indirect TENG devices with magnetic assistance were able to effectively harvest energy from water flow and wind to light up 50 white LEDs. 

In addition, Huang et al. looked at how the output performance was affected by the mass ratio, the distance between the magnet and the magnetic response layer, and the strength of the magnet [111]. The distance between the electrodes can be changed by increasing the magnet content, and the output performance of the TENG device can be improved by strengthening the magnetic field (Figure 7a). Furthermore, when the magnetic-assisted non-contact TENG was prepared with a mass ratio of 1:3 (PDMS/Fe-Co-Ni) and operated at a magnetic field strength of 5 kOe and frequency of 10 Hz, the Voc and Isc reached approximately 275 V and 9 µA, respectively (Figure 7b). This gadget has great promise for use in optoelectronics, self-powered electronics, magnetic sensing, and energy harvesting in the future. However, the magnetic field force may shorten the device’s lifespan and cause damage to the device.

Using a magnetic covalent organic framework hybrid (Fe_3_O_4_@COFs) as a favorable triboelectric material, Ghosh et al. created a COF TENG (CS-TENG) for the first time [112]. In this design, a PDMS film incorporated with the two-dimensional transition metal carbide/nitride (MXene) was used as the negatively charged triboelectric material, and the exterior surface of the PDMS/MXene film was optimized using 3D printing and etching techniques. Fe_3_O_4_ increases the surface charge density of the cathode material. The content of Fe_3_O_4_ in Fe_3_O_4_@COFs influenced the electrical energy output, with Fe_3_O_4_@COF-2 containing 300 mg showed approximately twice the performance compared to bare COF. At a contact frequency of 10 Hz and a contact force of 2.8 N, the CS-TENG produced a peak Voc of about 146 V and an Isc of 32 µA (Figure 7c,d). Furthermore, the alternating current (AC) electrical signal was rectified into direct current (DC) for water electrolysis, and the generation of hydrogen and oxygen was confirmed by scanning electrochemical microscopy (SECM) (Figure 7e). Consequently, reliance on solar energy and other non-renewable energy sources can be decreased by converting this green hydrogen generation technique employing additional renewable mechanical energy supplies. However, the capacity for the output of this TENG is insufficient, and the water electrolysis efficiency is slow.

To improve output performance and durability, Tang et al. reported a malleable Halbach magnetic array-assisted sliding mode triboelectric nanogenerator (MA-S TENG) that used rubber and strontium ferrite magnetic powder as substrates [113]. Periodic contact separation was produced from the sliding motion of the covered sliding component (Figure 7i). Moreover, under the same contact area, due to the different working modes, the MA-S TENG exhibited 2.7 times the output energy of an independent TENG in a single sliding action, and the peak power was over 10 times that of an independent TENG (Figure 7f,g). Additionally, the MA-S TENG kept 60 percent of its original output capability after 25 h of continuous operation, compared to 1.8% for the independent TENG. An efficient and promising method for enhancing TENGs’ output performance and durability is presented in this study.

Wang et al. created a micro-scale TENG based on shear thickening fluid (STF) and magnetically sensitive thin films (carbonyl iron (CI)) that has excellent structural durability, impact resistance, and the capacity to gather energy from compression, oscillation, and magnetic fields [57]. The ecoflex/CI-50% (magnetization:117.04 emu/g) TENG demonstrated a voltage of 10.40 V and a maximum power density of 27.05 mW/m^2^ when subjected to a compression load of 10 MΩ (Figure 7h). The TENG was also given magnetically programmable shapes through the high magneto-sensitive effect, which allowed it to detect and gather minute STF flow energy in a non-contact mode. Crucially, the STF equipped device showed remarkable protective capability for both the wearer and the TENG by withstanding and dissipating strong collision forces ranging from 1390 N to 409 N.

Firdous et al. propose doping Fe_3_O_4_ into PVDF (P/Fe_3_O_4_) by electrospinning as a way to increase a TENG’s output power. The interaction between fluoride and Fe drove the surface friction to become positively charged [114]. By adjusting the doping ratio, the dielectric constant of P/Fe_3_O_4_ increased, and the surface of P/Fe_3_O_4_ transitioned from positive to negative friction. Surface potential equalization and charge transfer channels were produced when an external magnetic field was supplied because the implanted negative centers reacted to the magnet and were biased toward the electrode. The charge transfer amount (Qsc) increased to 47 nC, generating a current of 4.8 µA at 84 MΩ (Figure 7j), with a power density of 1.66 W/m^2^ (Figure 7k). This indicates a 15,900% growth in Isc (0.03–4.8 µA). The design of this TENG holds great potential for high-power-demand electronic devices. 

**Table 3 nanomaterials-14-00826-t003:** Applications and properties of composite magnetic materials in TENG.

Material	Driving Mode	Key Materials	Output Voltage	Output Current	Output Power/ Power Density	Ref.
Strontium ferrite	Contact	PTFE/Strontium ferrite	0.6 V	—	—	[58]
Fe_3_O_4_	No contact	PVDF-Fe_3_O_4_	75 V	6 µA	0.23 mW	[109]
Fe_3_O_4_	Contact	CNTs@Fe_3_O_4_	—	1.1 µA	147.9 µW	[110]
Fe-Co-Ni	No contact	PDMS/Fe-Co-Ni	206 V	30 µA	3 mW	[59]
Fe-Co-Ni	No contact	PDMS/Fe-Co-Ni	275 V	9 µA	—	[111]
Fe_3_O_4_	Contact	Fe_3_O_4_@COFs/PDMS-MXene	146 V	32 µA	8.04 µW/m^2^	[112]
SrFe_12_O_19_	Contact	FEP-SrFe_12_O_19_/Al	495 V	130 µA	19.0 mW	[113]
Carbonyl iron	No contact/Contact	Ecoflex/CI	23.59 V	8.84 µA	27.05 mW/m^2^	[57]
Fe_3_O_4_	Contact	PVDF/Fe_3_O_4_	—	4.8 µA	1.66 W/m^2^	[114]

## 4. Materials in Self-Powered Sensors

By adding magnetic materials to different substrates, the ability to sense magnetic fields or enhance the performance of self-powered sensors can be achieved. Magnetic material-based sensors have been widely used in various applications. However, based on the characteristics of magnetic materials and the current research status of self-powered sensors, self-powered sensors using magnetic materials mainly include magnetic field, temperature, humidity, pressure, and other sensors (Table 4). In the following section, the application of magnetic materials in self-powered sensors will be introduced.

### 4.1. Self-Powered Magnetic Sensors

The inherent magnetism of magnetic materials provides advantages in terms of magnetism for self-powered sensors, which other materials do not possess. It can also enhance the magneto-electric coupling performance of the materials. By incorporating CI nanoparticles into the PVDF matrices, Sang et al. published in 2018 a multifunctional PVDF/CI magneto-electric composite film using a solution-casting process [115]. The composite film’s tensile strength and Young’s modulus were enhanced by the addition of magnetic CI nanoparticles and exhibited significant magneto-mechano-electric coupling characteristics, generating electrical signals when subjected to magnetic fields or bending deformation (Figure 8a). Among them, for bending displacements of 2, 4, 6, 8, and 10 mm, the output charge of the 10 wt%-PVDF/CI composite film was 3.0, 9.6, 14.9, 18.6, and 24.6 pC, in that order (Figure 8b). The magneto–electric charge produced by the 10 wt%-PVDF/CI composite film rose from 0 to 676 pC when the magnetic field changed from 0 mT to 600 mT (Figure 8c). Moreover, polynomial fitting can be utilized to identify the numerical relation (0.97) between the magneto–electric charge and the magnetic field. This relationship can be used to calculate the strength of the surrounding magnetic field. In addition, the PVDF/CI composite film exhibits good flexibility and stability, indicating its potential application value in shape-adaptive sensors and magnetic field-related sensors in the future.

Furthermore, Wang and colleagues synthesized magnetorheological elastomers (MRE) in 2020 by combining varying concentrations of CI nanoparticles into matrices of shear stiffening elastomer/PDMS (SSG/PDMS) [116]. With a strong magnetorheological (MR) effect of 114.68% and a maximum storage modulus of 0.77 MPa, the SSG/PDMS-60% containing 60% CI demonstrates the material’s fast reactivity to magnetic fields in terms of its rheological properties. Among them, the 4 mm thick TENG exhibits the highest energy harvesting efficiency, is able to power 39 LEDs with a maximum Isc of 2.62 mA (Figure 8d), and has an optimal output power of 55.07 mW (Figure 8e). Furthermore, the external magnetic field intensity can control the output performance of the TENG. The Voc of the TENG drops from 20.99 V to 1.81 V when external magnetic flux density increases from 0 to 190 mT (while keeping the compressive load at 60 N) (Figure 8f). The self-powered sensors fabricated using the MRE material can not only distinguish various mechanical movements but also sense the strength of magnetic fields. The addition of CI particles demonstrates the functionality of the TENG as a sensor for detecting external magnetic fields.

In 2021, Hajra et al. synthesized a rare earth magnetic material (1 − *x*)BaTiO_3_−*x*ErFeO_3_ (BTEFO) through a solid-state reaction for the fabrication of high-performance hybrid HNG devices (MF-HG) [117]. The doping of rare earth ferrite ErFeO_3_ into BaTiO_3_ does not affect its crystal lattice, thereby extending the material’s magnetic properties and enhancing its dielectric and piezoelectric performance. The combination of 3 weight percent ErFeO_3_ in BaTiO_3_ and 10 weight percent BTEFO in PDMS produced the best results among them (remnant polarization: 1.21 µC/cm^2^, maximum value of magnetization: 0.026 Gauss), with a Voc of 260 V and Isc of 3 µA. Furthermore, following surface treatment and multi-unit integration, 8.5 µA of current, 0.25 W/m^2^ of power density, and 320 V (Figure 8g) of voltage were attained. Figure 8h,i shows that the output also changes with varying magnetic field intensities. This suggests that the multi-unit MF-HG can function as a sensor for detecting external magnetic fields and has a wide range of potential uses in defense, magnetic induction, and magnetic mineral exploration. 

### 4.2. Self-Powered Temperature and Humidity Sensors

Peng et al. proposed a magnetic soft millirobot, modeled after a slug, fully integrated with a TENG (TENG-Robot) [61]. The TENG-Robot is integrated onto a magnetic sheet composed of NdFeB particles mixed with PDMS in a weight ratio of 4:1, measuring 50 × 20 mm^2^ and 0.93 mm in thickness (Figure 9a). The 0.4 mm thick magnetic sheet (magnetic flux intensity: 75 mT) provides mechanical energy through the periodic motion of contact separation under the action of a magnetic field. The TENG-Robot can concurrently generate frictional charges in a CS while moving on flat surfaces, hills, and gaps by varying the direction of the magnetic field. The TENG-Robot can crawl at 25 cm/s (at a frequency of 20 Hz), turn at 120 m/s (at a frequency of 16 Hz), and move on a 60° inclined paper slope (at a frequency of 12 Hz) at a magnetic field strength of 70 mT. At a frequency of 10 Hz, it can produce a peak Voc of 120 V and a peak Isc of 10 µA (Figure 9b). Additionally, the TENG-Robot incorporates energy storage capabilities and the ability to sense external temperature and emit light.

Based on Maxwell’s equations and micromagnetic theory, Liu et al. confirmed that linking the magnetization current and displacement current improved TENG production performance [118]. The assistance of magnetic materials can increase the magnetization current during periodic contact friction. Therefore, a PDMS/Fe (PICF) composite film was prepared by adding Fe as an auxiliary material. The initial magnetization strength of PICF under externally applied positive and negative magnetic fields throughout the PICF fabrication process, as well as the link between the concentration of Fe particles in the PDMS/Fe combination and TENG output, were examined. The Isc density of the TENG was 27 mA/m^2^ and the immediate electrical density was 2 W/m^2^ (Figure 9d) when the Fe concentration was 2 wt% and the applied magnetic field intensity was 20 mT. These values were, respectively, 800% and 8100% much greater than that of the TENG prepared with pure PDMS (Figure 9c). Additionally, by assembling 16 units into a TENG array, dispersed energy can be effectively collected for temperature and humidity sensing systems (Figure 9e). The study of effective energy-collecting devices, flexible electronics, self-powered sensors, and protective electromagnetic equipment is supported by the magnetically assisted TENG.

Tayyab et al. synthesized PVDF-PI NFs with different growth times by electrospinning printer ink (PI) with magnetic particles and PVDF nanofibers (NFs), and used them to prepare a humidity-responsive TENG [63]. The magnetic particles improved the output performance of the TENG by increasing the surface charge density of the material. The PVDF-PI NFs (saturation magnetization: 0.015 emu/cm^2^) that were grown for the greatest period of time—five hours—showed the highest peak power of 22 W/m^2^, far surpassing the PVDF NF material that was grown for the same amount of time (Figure 9f). The collaboration of the PVDF NFs’ dipoles and PI nanofillers was primarily responsible for this improvement. Therefore, the PVDF-PI NF material can be efficiently applied in flexible wearable devices and self-powered sensor devices. This study not only provides inspiration for the use of nanoparticles in PVDF NFs, but also contributes to the further investigation of magnetic, dielectric, and strain properties of self-powered systems.

Xiang et al. used graphene oxide (GO), carboxylated chitosan, and magnetic Fe_3_O_4_ to create a flexible magnetic nanoparticle–GO–carboxylated chitosan composite sheet (MGC) [119]. They further prepared a magnetic M-TENG by incorporating MGC with different magnetic particle contents into polytetrafluoroethylene (PTFE) material. Magnetic particles (saturation magnetization: 29.4 emu/g) improved the output performance by increasing the friction between layers. The MGC exhibited excellent biocompatibility, with an ideal open Voc of 168.2 V and an Isc of 7.6 µA (Figure 9g). The highest power density reached 107.5 mW/m^2^ (Figure 9h), enabling rapid charging of a 47µF capacitor to 3.3 V while maintaining good stability. Additionally, the M-TENG demonstrated high sensitivity to different humidity levels (Figure 9i), making it suitable for self-powered humidity sensors and powering other electronic devices. In addition to its possible use in tracking human movement, the M-TENG is crucial for the observation and conservation of cultural artifacts. 

### 4.3. Self-Powered Pressure Sensors

Using polyethylene terephthalate (PET) and magnetite (Fe_2_O_3_) as the frictional layer material, Fatma et al. created a TENG in 2019 (Figure 10a) [120]. They investigated the effects of different Fe_2_O_3_ contents, frequencies, and pressures on the output capability of the TENG. The incorporation of magnetic particles increased the output performance by enhancing the surface charge density through the increased roughness. The TENG, comprising 15 wt% Fe_2_O_3_/PVDF (saturation magnetization: 6.25 emu/g, coercive field: 278.3 Oe), achieved an open-circuit voltage of 250 V, a short-circuit current of 5 µA (Figure 10b), and a power density of 0.117 W/m^2^. It also reached a maximum peak power of 0.17 mW, which was 5.6 times greater than that of the pure PVDF material. With a simple finger tap, the TENG device was capable of powering 108 series-connected LEDs continuously and recharging a 1 µF capacitor to 30 V within 90 s (Figure 10c), demonstrating its potential for self-powered wearable and portable electronic devices. Furthermore, the TENG exhibited remarkable sensitivity to slight variations in force and frequency, indicating its possible use as a self-contained pressure sensor. By operating in an indirect method, the magnetically responsive TENG prevented damage and lengthened its lifespan.

Furthermore, Wu et al. used physical vapor deposition to create a self-powered multifunctional sensor (MS) that can detect rotation parameters, forces, and accelerations by taking inspiration from the motion of a magnetic cylinder [121]. The TENG comprised a low-damping magnetic column (NdFeB) and a PTFE film. This MS generated voltage output across the electrodes by converting the translational movement of the magnetic column surrounding the PTFE film into swinging or multi-turn rotary movement (Figure 10d). The distance of the magnet could be adjusted to fine-tune the sensitivity and resolution of the MS. To ascertain force and acceleration parameters, it is also necessary to extract the amplitude, frequency, and certain temporal features from the output waveform (Figure 10e,f). Furthermore, a program was developed based on the MS to protect vehicle safety, triggering an alarm response when a stranger lightly touched the vehicle handle. Therefore, the MS showed great potential for applications requiring feedback on acceleration, force, and rotation parameters, such as robotics and mechanical control.

In 2021, Seo and collaborators synthesized a single-electrode mode TENG (C-TENG) with cilia microstructures using a magnetically guided method with polydimethylsiloxane–iron carbonyl (PDMS-Fe) (Figure 10g) [122]. By comparing the surface structures and output performance of PDMS-Fe composite materials with different Fe contents, it was found that PDMS-Fe with 10 wt% exhibited excellent output results. Magnetic particles improve the output performance by increasing the roughness. The C-TENG operated in a SE, and the Voc of the C-TENG synthesized with 10 wt% PDMS-Fe could reach 70 V, with a maximum Isc of 250 nA. With a 30 MΩ load, the maximum power density was 2.75 µW/cm^2^ (Figure 10h). Additionally, the C-TENG could convert mechanical energy such as wind energy and pressure into electrical energy output. As shown in the figure, it could sense objects of different weights and output corresponding voltages (Figure 10i), indicating potential applications in pressure sensors. Furthermore, after being integrated with a bridge rectifier circuit, the C-TENG could power capacitors and calculators, demonstrating that the addition of ions and microstructures could improve the self-powered output performance of the TENG. 

### 4.4. Other Self-Powered Sensors

Wan et al. created a flexible hybrid electromagnetic-TENG for energy harvesting and three-dimensional trajectory sensing by combining multi-walled carbon nanotubes (MWCNTs), PDMS, and NdFeB particles [123]. The flexible material, PDMS with magnetic and conductive properties (MC-PDMS), was used as the electrode for the TENG and as the magnet for the EMG. Increasing the NdFeB content can improve the magnetic flux to enhance the output performance. The TENG’s highest peak voltage was 103 V, and its maximum output current was 7.6 µA. Its maximum power density was 7.3 µW/cm^2^ (Figure 11a). The hybrid system could recharge a 10 µF capacitor to 3 V in 110 s (Figure 11c), demonstrating improved efficiency compared to using a single TENG/EMG for capacitor charging. Additionally, by attaching the MC-PDMS material to objects and utilizing a copper coil array and electromagnetic induction principle, the three-dimensional trajectory data for objects above the array could be obtained simply (Figure 11b). This HNG holds great potential for applications in flexible electronic devices and robotics.

Sun et al. utilized a composite film of magnetic nickel particles and silicone rubber as the friction material for the TENG to enhance its output performance [124]. The output performance of the TENG was significantly influenced by the composition and size of the magnetic particles. Under the action of the magnetic field, the capacitance of a certain concentration of nickel particles would increase due to the increase in the solidified magnetic flux density, improving the output performance. However, too high a concentration of nickel particles will leak due to a reduction in the gap. The highest peak voltage and current of the M-TENG were 233.4 V and 32.6 µA, respectively, when the concentration of nickel particles was 1 wt% and their particle size was 0.8 µm (Figure 11d). These values were 2.8 and 3 times higher than those of the TENG without nickel particles (O-TENG). With a 3 MΩ load resistor, the maximum output power of the M-TENG (2.5 mW) was 4.7 times higher than that of the O-TENG. In practical performance tests, a 100 μF capacitor could be charged to 3 V within 308 s (Figure 11e), and it could repeatedly light up 100 series-connected LED lights. Additionally, a sensing system was developed using the M-TENG, which automatically displayed target numbers (Figure 11f). This work extended the TENG’s use in autonomous sensing and enhanced its output performance from a fresh angle.

Nazar et al. utilized 3D printing to design a magnetic levitation-based TENG (ML-TENG) for energy harvesting and self-powered speed sensing systems (Figure 11g) [125]. The energy was collected through the contact of the magnetic capsules during the sliding mode using the mutual repulsive force of magnets. With the highest voltage output and power of 4 V and 340 μW, respectively, the copper ML-TENG in mode 2 outperformed the other modes in terms of output performance (Figure 11h). The ML-TENG was used to construct a self-powered car speed-detecting system, and testing was performed to determine how well the system performed at various vehicle speeds. At a speed of 15 km/h, it was discovered that the maximum output voltage was 7.2 V (Figure 11i). Furthermore, by helping to return the magnetic system to its initial condition, the magnetic materials improved the TENG’s endurance and made it easier to use self-powered speed monitoring devices. 

**Table 4 nanomaterials-14-00826-t004:** Applications and properties of magnetic materials in TENG sensors.

Materials	Driving Mode	Key Materials	Output 1	Power/ Power Density	Cycle Performance	Applications	Ref.
Carbonyl iron	Contact	PVDF/CI	—	676 pC (electric charges)	100% at 17 cycles	Magnetic sensor	[115]
Carbonyl iron	Contact	SSG/PDMS	20.99 V/2.62 mA	55.07 mW	20.99 V to 20.33 V after 500 cycles	Magnetic sensor	[116]
ErFeO_3_	Contact	BaTiO_3_/ErFeO_3_	320 V/8.5 µA	0.25 W/m^2^	—	Magnetic sensor	[117]
NdFeB	Contact	PDMS/NdFeB	120 V/10 µA	450 mW/m^2^	100% at 6000 cycles	Temperature sensor	[61]
Iron	Contact	PDMS/Fe	27 mA/m^2^	2 W/m^2^	100% at 6000 cycles	Temperature and humidity sensor	[118]
Fe_3_O_4_	Contact	PVDF-PI NFs	1600 V/130 μA	22 W/m^2^	100% at 1000 cycles	Humidity sensor	[63]
Fe_3_O_4_	Contact	MGC	168.2 V/7.6 μA	107.5 mW/m^2^	100% at 2400 cycles	Humidity sensor	[119]
Fe_2_O_3_	Contact/No contact	PVDF/PET	250 V/5 μA	0.17 mW/0.117 W/m^2^	100% at 3000 s	Pressure sensor	[120]
Ni/NdFeB-N38	Contact	PTFE	1.75 V/70 nA	1.05 nC (electric charge)	—	Force and acceleration sensor	[121]
Carbonyl iron	Contact	PDMS-Fe	70 V/250 nA	2.75 µW/cm^2^	100% at 1600 cycles	Pressure sensor	[122]
NdFeB	Contact	MC-PDMS	103 V/7.6 μA	7.3 µW/cm^2^	100% at 14,000 cycles	Three-dimensional trajectory sensor	[123]
Nickel particles	Contact	Cu/Al	233.4 V/32.6 µA	2.5 mW	—	Automatic target-scoring system	[124]
Magnet (M1)	Contact	Cu/Al	4 V	340 µW	—	Speed sensor	[125]

## 5. Conclusions and Outlook

This paper summarizes the applications of magnetic materials in TENGs and their applications in HNGs and self-powered sensors. It can be found that magnetic materials have shown great potential for developing high-performance and multifunctional TENGs for flexible wearables and robotics. Furthermore, composite films formed by magnetic particles and other materials, combined with magnets, enable non-contact energy harvesting, thereby improving the stability and durability of TENGs in extreme environments. Additionally, mechanical information in the environment can be converted into output signals of TENGs, allowing TENGs to operate as independent sensing systems and aid in the quick advancement of the Internet of Things era. 

While magnetic material-based self-powered sensors offer many advantages, there are certain limitations in terms of integration and practical applications.

### 5.1. Power Output Enhancement

The methods to boost the output performance of TENGs mainly include material selection, surface optimization, and structural design. In this review, magnetic materials are chosen, which can be added as tiny magnetic particles to the electrode materials (with an excellent surface charge density, electronegativity, dielectric constant, etc.) or applied as thin films in the devices to increase the output performance of TENG through electrostatic induction and electromagnetic induction. Surface optimization is mainly achieved by physical or chemical treatments on the material surface. For example, changing the texture and roughness of the surface through physical methods such as electrospinning, 3D printing, and plasma treatment can improve the energy conversion efficiency. Chemical methods, such as surface corrosion or functionalization, can promote the transfer of electrons during the friction process to enhance the TENG’s output performance. Optimizing the structural design can also improve the output performance of soft or hard magnetic TENGs in different environments. By designing hybrid generators of different magnetic TENG structures (EMGs and TENGs), the coupling effect between triboelectrification and electromagnetic induction can be enhanced, enabling more efficient energy collection and conversion into electricity in a magnetic field. 

In addition, based on Maxwell’s equations, the addition of magnetic materials enhances the output current in TENGs. In the absence of an external magnetic field, the magnetic material undergoes magnetization due to the electric current generated after periodic friction with dielectric materials, thereby producing a magnetization current to enhance the output performance of the TENG. Upon the application of an external magnetic field, the magnetic material exhibits ordered magnetic domains, leading to a stronger magnetization current, while the magnetic field’s intensity directly impacts the magnitude of this current. Currently, there is relatively limited research on the enhancement of TENG performance through the magnetization current.

While all of these techniques can enhance a magnetic TENGs’ output performance, there is still potential to maximize output performance through structural design.

### 5.2. Durability Enhancement

Magnetic materials possess strength and wear resistance, allowing them to maintain their magnetic properties and structural stability even after multiple stress and friction cycles, thereby improving the reliability and lifespan of TENGs. Durability is an important parameter in practical applications of TENGs. Frictional materials may experience certain wear during long-term bending and stretching, so improving the stability of TENGs requires considering the material’s wear resistance. The repulsion between magnets can also enhance the material’s fatigue resistance. Additionally, in different environments, temperature, humidity, ion concentration, and physical fields can reduce the durability of TENGs. Therefore, materials with certain resistance capabilities and non-contact encapsulation of TENGs can be considered. Magnetic materials can be used to control enclosed TENGs through a magnetic field, converting mechanical energy into electrical energy. Enclosed TENGs operate under constant humidity and ion concentrations, allowing non-contact TENGs to provide a stable output of electrical energy.

### 5.3. Multifunctionality

Multifunctionality holds significant value in practical applications, as it can boost the output performance of TENGs in various ways. TENG sensors with multifunctionality can also analyze complex environments and obtain multiple sets of data. Magnetic materials not only serve as critical components of TENGs but also can be used to secure and guide non-magnetic materials, thereby increasing the efficiency of mechanical energy conversion. Furthermore, magnetic materials can regulate the output voltage and frequency of the generator by altering their physical properties, thereby controlling the power generation performance.

The self-powered multifunctional sensor (MS) indicated above has the ability to sense acceleration, rotational parameters, and force. This TENG consists of a low-damping magnetic column and a PTFE film. The mechanism drives the electrodes to produce voltage output by converting translational motion into multiturn or swinging rotational movement of the magnetic column surrounding the PTFE film. Parameters of force and acceleration can be determined by extracting the amplitude, frequency, and some time characteristics from the output waveform. In the sliding mode, these miniature devices can be applied to equipment such as robots, drones, and aircraft that require motion parameters, thereby enhancing the sensitivity of the devices. This represents a significant advancement in this field.

### 5.4. Expansion of Application Scenarios

Magnetic materials possess magnetic properties that are not found in other materials, and there is a close relationship between magnetism and electricity. Therefore, by employing rational structural design, magnetic materials can not only enhance the output performance of TENGs but also boost their ability to sense magnetic fields. This has led to the development of self-powered sensors capable of sensing magnetic fields and magnetic minerals. Additionally, different magnetic materials can provide further assistance to TENGs and sensors in a variety of domains, such as energy, biology, medicine, flexible electronics, and the IoT, offering significant research opportunities. It is hoped that this summary will offer readers a thorough grasp of the present magnetic TENGs and act as a source of inspiration for robotics and internet research in the future.

## Figures and Tables

**Figure 1 nanomaterials-14-00826-f001:**
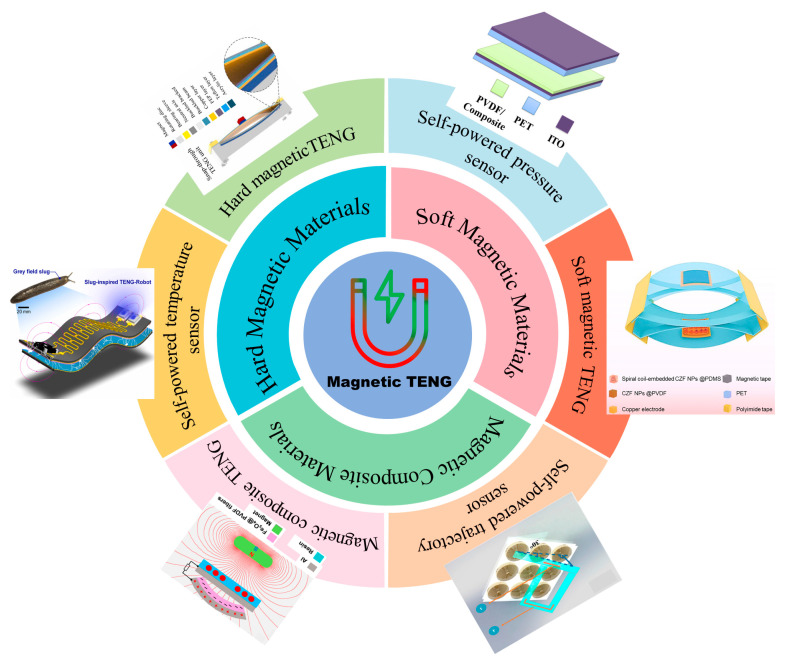
Progress of TENGs with magnetic materials.

**Figure 2 nanomaterials-14-00826-f002:**
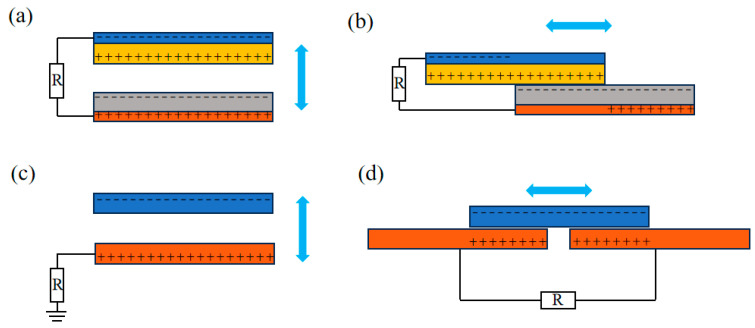
TENG operating modes: (**a**) vertical contact-separation mode, (**b**) lateral sliding mode, (**c**) single-electrode mode, and (**d**) freestanding triboelectric-layer mode.

**Figure 3 nanomaterials-14-00826-f003:**
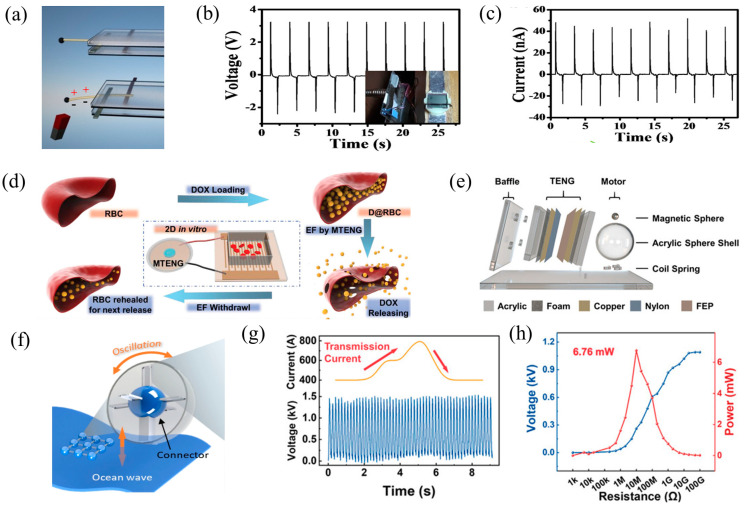
(**a**) Magnetic force-driven contactless nanogenerator (CLNG); the ICLNG’s output voltage (**b**) and current (**c**) are made up of five PZT nanowire array pieces connected in series, 2012 ACS [94]. (**d**) Diagram illustrating the process of loading DOX into RBCs and then integrating an MTENG to achieve the regulated release of DOX, 2019 Wiley [67]. (**e**) Schematic structure of RB-TENG, 2021 Wiley [95]. (**f**) The MC-TENG applied to ocean waves, 2022 Springer Nature [96]. (**g**) The RB-TENG voltage curve during a high current shock; (**h**) profiles of the output power resistance, 2021 Wiley [95].

**Figure 4 nanomaterials-14-00826-f004:**
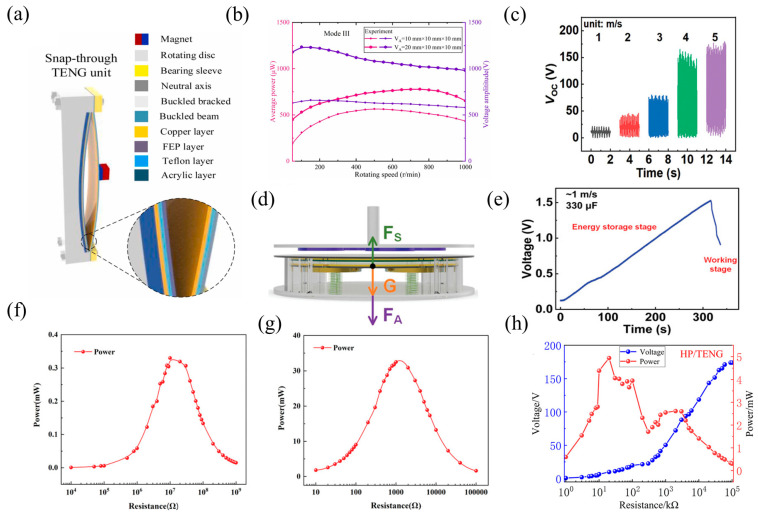
(**a**) Snap-through triboelectric nanogenerator (ST-TENG) unit. (**b**) The impact of the stationary magnet on the average power and voltage amplitude of ST-RTENG, 2022 Elsevier [16]. (**c**) AMF-TENG’s Voc at various wind speeds, (**d**) force analysis of the vibrator, and (**e**) a 330 µF commercial capacitor charged at 1 m/s for 316 s with the AMF-TENG is charged to 1.5 V, 2023 Wiley [55]. (**f**,**g**) At 400 rpm rotation, the median output power is under varying external load resistance, 2021 Wiley [56]. (**h**) The HP/TENG’s output voltage and power depend on the resistance of the external load, 2021 Wiley [97].

**Figure 5 nanomaterials-14-00826-f005:**
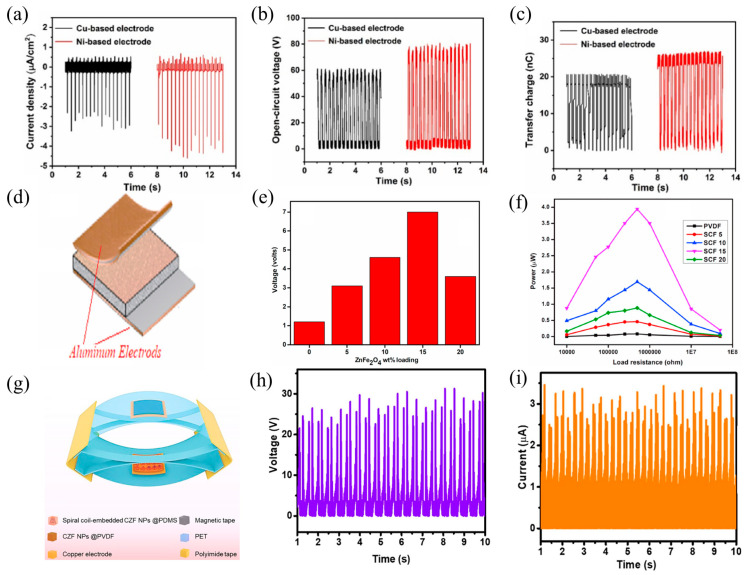
The electrodeposited Cu, Ni-based TENG’s short-circuit current density (**a**), open-circuit voltage (**b**), and transferred charges (**c**), 2021 Elsevier [62]. (**d**) Schematic presentation of a nanogenerator, 2022 Elsevier [103]. (**e**) Voltage response histogram as a function of ZnFe_2_O_4_ loading, and (**f**) power of composite PVDF and PVDF/ZnFe_2_O_4_ nanogenerators with varying load resistances, 2021 Elsevier [104]. (**g**) HNG in three dimensions, with the combined output voltage (**h**) and current (**i**) represented, 2022 Elsevier [105].

**Figure 6 nanomaterials-14-00826-f006:**
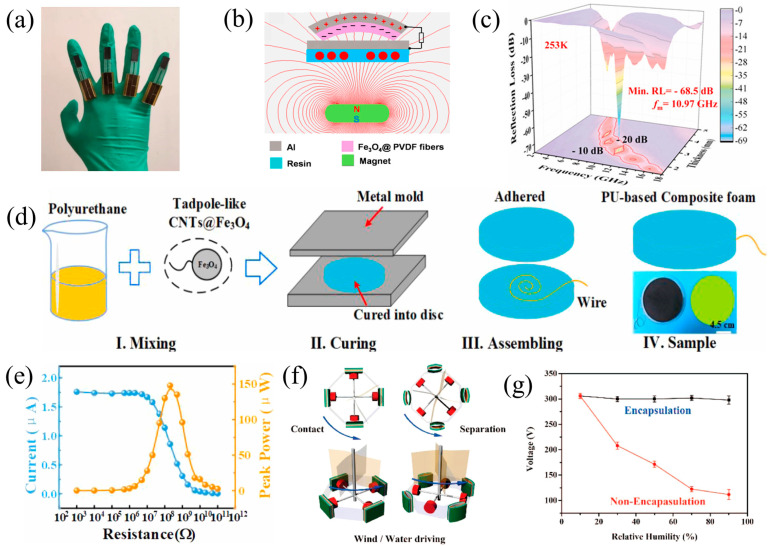
(**a**) Triboelectric sensor worn on the fingers, 2021 Springer Nature [58]. (**b**) Schematic diagrams of the hybrid EM-TE nanogenerator, 2017 Elsevier [109]. (**c**) The 90CNT@Fe_3_O_4_/PU CF-TENG reflection loss (RL) at a temperature of 253 K as a function of frequency and sample thickness, as well as the lowest RL values; (**d**) an illustration of the CF-TENG fabrication scheme, (**e**) peak power, and current at different resistances, 2022 Elsevier [110]. (**f**) Schematic representation of a noncontact TENG with magnetic assistance for capturing wind and water movement energy, and (**g**) a comparison of the performance of a single noncontact TENG assisted by magnetism with and without enclosure, 2016 Elsevier [59].

**Figure 7 nanomaterials-14-00826-f007:**
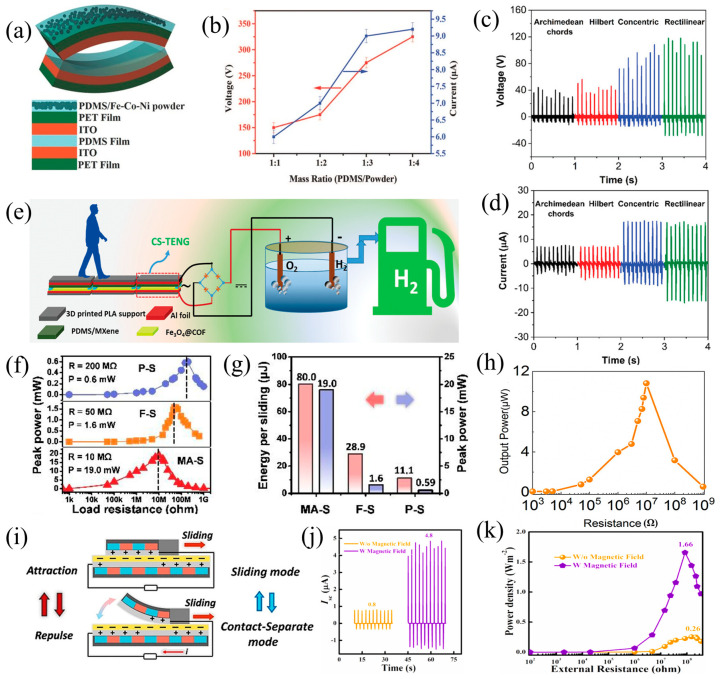
(**a**) Diagrammatic representation of the triboelectric generator (**b**) comparing the device’s voltage and short-circuit current to its mass ratio, 2016 Wiley [111]. PDMS/MXene sheets with varying etched patterns for the (**c**) Voc and (**d**) Isc, and (**e**) the distance between contacts. Biomechanical energy is transformed into green hydrogen fuel by the TENG, 2023 Wiley [112]. (**f**) Peak power under three distinct modes’ varying load resistances, and (**g**) the maximum output energy per sliding cycle (1 Hz), 2019 Elsevier [113]. (**h**) The output power of the TENG at different resistances, 2020 Elsevier [57]. (**i**) Diagrammatic representation of the MA-S TENG sliding process, 2019 Elsevier [113]. Comparison of the device’s output capability in the presence and absence of a magnetic field; (**j**) Isc and (**k**) power density under a resistive load, 2021 Elsevier [114].

**Figure 8 nanomaterials-14-00826-f008:**
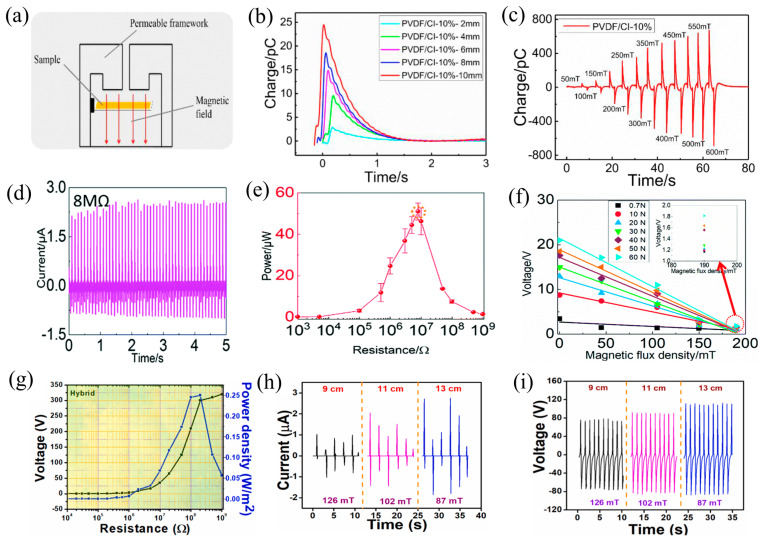
(**a**) Diagram for a single-way bending test in a magnetic field, (**b**) charge amounts of 10%-PVDF/CI films at different bending displacements, and (**c**) the amount of charge of 10%-PVDF/CI magnetic films at different magnetic field strengths, 2018 Elsevier [115]. (**d**) Output current at 8 MΩ, (**e**) output power at various external resistance values at load and a frequency of 60 N and 10 Hz, and (**f**) the correlation between the magnetic flux density and voltage at various pressures, 2020 Royal Society of Chemistry [116]. (**g**) Power density and voltage at various resistors, and (**h**,**i**) the effect of the distance between magnets and periods on the current and voltage, 2021 Elsevier [117].

**Figure 9 nanomaterials-14-00826-f009:**
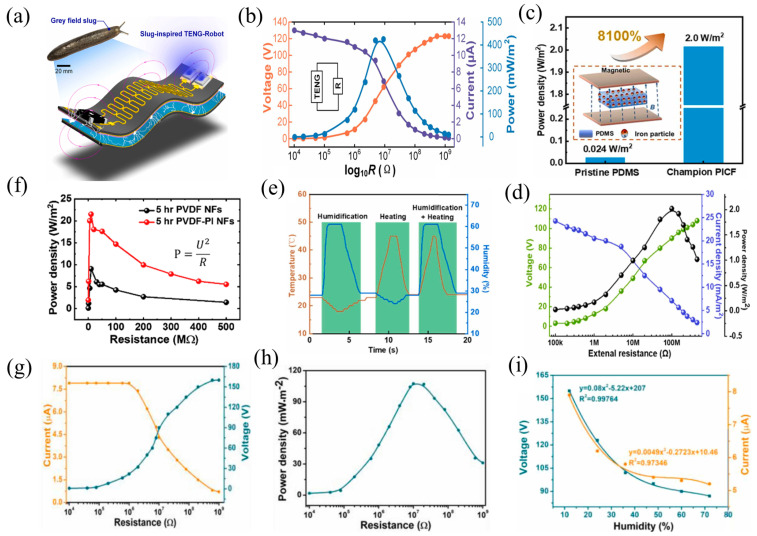
(**a**) Diagram of the TENG-Robot and (**b**) the impact of an external load impedance on the TENG-Robot’s electrical output performance, 2022 Elsevier [61]. (**c**) The power density of PICF and pure PDMS materials, (**d**) the TENG’s instantaneous power density, output voltage, and current at various resistances, and (**e**) the signal of the self-powered humidity–temperature sensing system, 2022 Elsevier [118]. (**f**) The two materials’ power densities after five hours of growth, 2020 Elsevier [63]. The M-TENG’s output performance (Voc, Isc (**g**), and power densities (**h**)) while connecting various resistances, and the (**i**) fits of the curves for the Voc and Isc relationship for the M-TENG at varying humidity levels, 2022 Elsevier [119].

**Figure 10 nanomaterials-14-00826-f010:**
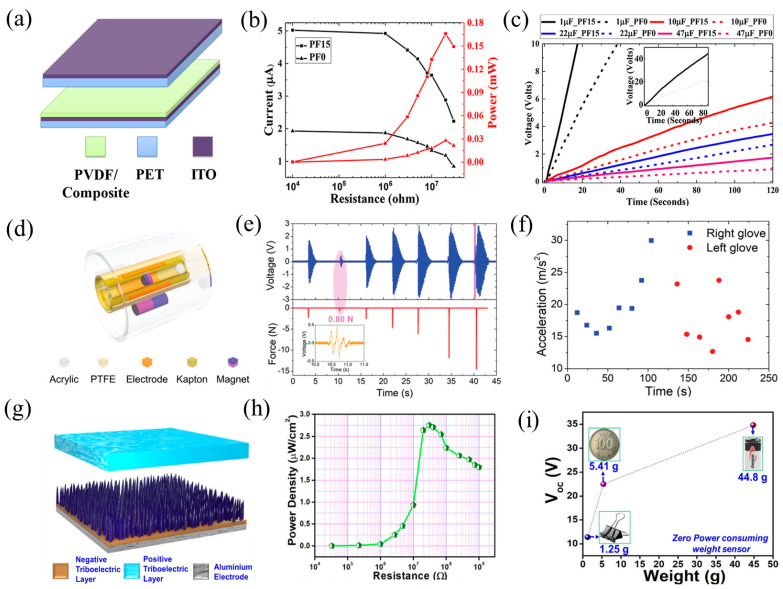
(**a**) TENG device structure. (**b**) For both pure PVDF and PVDF with magnetic nanoparticles, the external load resistance affects the output current and power density; (**c**) voltage curves for charging pure PVDF and 15% weighted magnetic nanocomposite capacitors, 2019 ACS [120]. (**d**) Schematic illustration of the MS, (**e**) findings that match up with the impact force and the MS’s Voc, and (**f**) acceleration test results for the right and left gloves, 2019 Wiley [121]. (**g**) TENG based on cilia, (**h**) power density at different loads, and (**i**) TENG self-powered load cell based on 10 wt% Fe-PDMS, 2021 Elsevier [122].

**Figure 11 nanomaterials-14-00826-f011:**
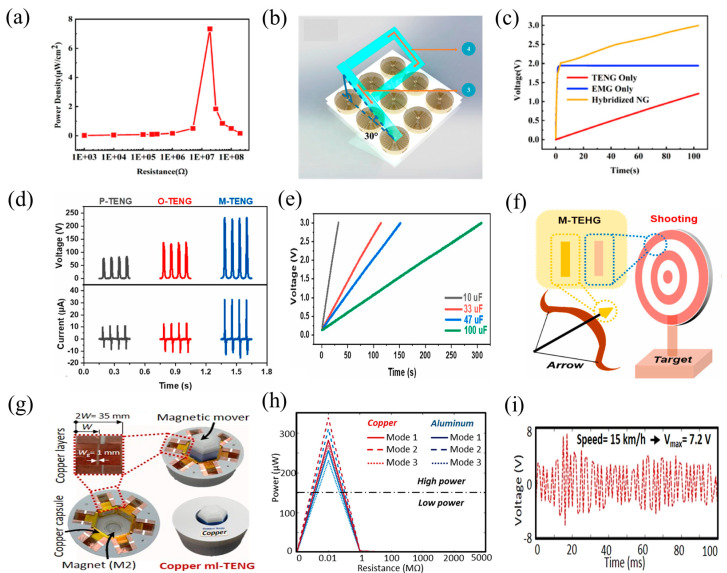
(**a**) The instantaneous power density of TENG, (**b**) writing the letter “P” at a rather steady rate above the array in three dimensions, and (**c**) 10 μF capacitance charging using the TENG, EMG, and HNG, 2020 Elsevier [123]. (**d**) Voc and Isc of three kinds of TENGs, (**e**) the M-TENG’s charging capacity with a film that was cured at 0.6 T and had 1 weight percent nickel particles measuring 0.8 μm, and (**f**) a schematic diagram of automatic target firing, 2020 Elsevier [124]. (**g**) Specifics of the ML-TENG’s copper layers, (**h**) the ML-TENG’s power distributions with respect to electrical resistance, and (**i**) voltage distributions derived from the ML-TENG under 15 km/h, 2021 AIP Publishing [125].

## Data Availability

Data are available upon request from the authors.

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
