# Peer review of "Magnetic Material in Triboelectric Nanogenerators: A Review"

_nanomaterials, 2024, doi:10.3390/nano14100826_

Round 1

Reviewer 1 Report

Comments and Suggestions for Authors

The authors have submitted a manuscript entitled “Magnetic Material in Triboelectric Nanogenerator: A Review”. The manuscript may not be suitable for publication as it does not focus or address any specific problem or outline. However, I would like to provide a few suggestions.

The reference number 94 is discussed under the section of hard magnetic materials. The authors clearly explain what kind of hard magnetic materials are used there. Similarly, all other references discussed in the same section. The authors did not mention what are the hard magnetic materials used in that study and how those hard magnetic materials are helping to enhance the performance of TENG.

The authors may provide a table with a list of hard magnetic materials used and TENG and their performance.

There is not much discussion of the soft magnetic materials and their application in TENG. Authors may refer to a few more articles for the soft magnetic materials and their performance. The authors may include a table listing the utilization of soft magnetic materials and their performance.

Similarly, the authors may provide a table for the composite materials in section 3.3.

In most places, acronyms do not have abbreviations. The authors should introduce the acronyms when they are used for the first time.

In some places, the authors have discussed multiferroic materials, it can be avoided and focused only on soft and hard magnetic materials. For example, reference 115.

In section 4, the authors randomly discussed various materials of sensors. There is no focus or proper outline on this paper.

There is no discussion of the magnetic properties such as saturation magnetization and coercivity of the magnetic materials used in this article. The authors should include these values of each material in each section.

The whole manuscript may have content focusing on the soft and hard magnetic materials in TENG.

Reviewer 2 Report

Comments and Suggestions for Authors

The review presented by Enqi Sun et al. summarizes the recent progress of applications of magnetic materials in the design and operation principle of triboelectric nanogenerators (TENGs). The review adheres to the journal’s standards, and has an appropriate figures appearance, but requires to be improved via eliminating the imperfections, for example:

1.     The main one is the state in the introduction section, where a clear idea of why your review is unique and presents information better than the others.

2.     One of the important issues lacking in this review is the summary of the more common materials available at the moment in the area of TENG. This can be presented either in a Table or as an additional paragraph.

3.     The quality of the figures is very low.

Comments on the Quality of English Language

Minor editing of English language required

Reviewer 3 Report

Comments and Suggestions for Authors

In the review, the authors summarize the possibilities of using magnetic materials in triboelectric nanogenerators of heating elements. The review discusses the basic principles of operation of heating elements, and also analyzes the latest achievements in the field application of magnetic materials in the development of triboelectric nanogenerators dividing them into soft ferrites, amorphous and nanocrystalline alloys. The problems and prospects of magnetic materials are discussed, as well as future opportunities for increasing their efficiency in energy conversion., the most promising options already available today and new applications are considered.The review will be useful to a wide range of readers.

Reviewer 4 Report

Comments and Suggestions for Authors

In this manuscript  the authors present a review about the magnetic material used in triboelectric nanogenerator.  They present basic principles of the operating modes of TENG then showing the recent progress in the use of magnetic materials to develop TENG highlighting the challenges and perspectives

In my opinion the manuscript document is well organized but in my opinion the paper needs some minor revisions as following reported:

- the authors presented the TENG with details but at the same time in my opinion the authors have not present in the same way (deeply) the magnetic material during the whale manuscript. In my opinion the authors should add a table that summarize the TENG different devices and then all the magnetic materials evolved in the development of TENG.

-the authors affirm that “…TENGs offer a more sustainable and eco-friendly … line 47”. In my opinion this statement needs to be supported by some references that compare the sustainability of the whole development and end-life cycle of TENGs and the materials used to develop it vs other energy harvesting technologies.

-the authors well introduced the TENG operative modes “line 117” but after this paragraph the authors should discuss the modes in the order in which they have been presented (SE, LS, ...)

-line 166: the authors introduce magnetic material such as NdFeB , in my opinion the authors should add a detail of such materials. In my opinion the authors, at this point, did not highlighted  the magnetic materials importance !!!! and the same issue occurs in the whole manuscript.

-in the section 4.2 “Self-Powered Temperature and Humidity Sensors” , the authors did not report any example of humidity sensors … or  discussion about it …..please provides same comments.

-in additions are there any applications in chemicals sensors field ?  please add some comments or section in the manuscript.

-in line 716 the authors report different power output enhancement, in my opinion the authors should add some example and references about the 3D printing  technology in this field of generators development.

Round 2

Reviewer 1 Report

Comments and Suggestions for Authors

The revised manuscript may be considered for publication